# Controlling Air Bubble Formation Using Hydrophilic Microfiltration Diffuser for *C. vulgaris* Cultivation

**DOI:** 10.3390/membranes12040414

**Published:** 2022-04-11

**Authors:** Siti Nur Alwani Shafie, Wong Yoong Shen, Jc Jcy Jaymon, Nik Abdul Hadi Md Nordin, Abdelslam Elsir Elsiddig Mohamednour, Muhammad Roil Bilad, Lam Man Kee, Takeshi Matsuura, Mohd Hafiz Dzarfan Othman, Juhana Jaafar, Ahmad Fauzi Ismail

**Affiliations:** 1Department of Chemical Engineering, Universiti Teknologi PETRONAS, Seri Iskandar 32610, Perak Darul Ridzuan, Malaysia; alwanishafie93@gmail.com (S.N.A.S.); yoongshen@hotmail.com (W.Y.S.); jc.jcy25330@utp.edu.my (J.J.J.); abdelslam20001785@utp.edu.my (A.E.E.M.); lam.mankee@utp.edu.my (L.M.K.); 2Department of Chemical and Process Engineering, Universiti Brunei Darussalam, Jalan Tungku Link, Gadong BE1410, Brunei; roil.bilad@ubd.edu.bn; 3Department of Chemical Engineering, University of Ottawa, 75 Laurier Ave. E, Ottawa, ON K1N 6N5, Canada; matsuura@eng.uottawa.ca; 4Department of Chemical Engineering, Universiti Teknologi Malaysia, Johor Bahru 81310, Johor, Malaysia; dzarfan@utm.my (M.H.D.O.); juhana@utm.my (J.J.); afauzi@utm.my (A.F.I.)

**Keywords:** membrane bubble diffuser, hydrophilic coating, bubble formation

## Abstract

In this project, a commercial polytetrafluoroethylene (PTFE) membrane was coated with a thin layer of polyether block amide (PEBAX) via vacuum filtration to improve hydrophilicity and to study the bubble formation. Two parameters, namely PEBAX concentration (of 0–1.5 wt%) and air flow rate (of 0.1–50 mL/s), were varied and their effects on the bubble size formation were investigated. The results show that the PEBAX coating reduced the minimum membrane pore size from 0.46 μm without coating (hereafter called PEBAX0) to 0.25 μm for the membrane coated with 1.5wt% of PEBAX (hereafter called PEBAX1.5). The presence of polar functional groups (N-H and C=O) in PEBAX greatly improved the membrane hydrophilicity from 118° for PEBAX0 to 43.66° for PEBAX1.5. At an air flow rate of 43 mL/s, the equivalent bubble diameter size decreased from 2.71 ± 0.14 cm for PEBAX0 to 1.51 ± 0.02 cm for PEBAX1.5. At the same air flow rate, the frequency of bubble formation increased six times while the effective gas–liquid contact area increased from 47.96 cm^2^/s to 85.6 cm^2^/s. The improved growth of *C. vulgaris* from 0.6 g/L to 1.3 g/L for PEBAX1.5 also shows the potential of the PEBAX surface coating porous membrane as an air sparger.

## 1. Introduction

Dissolving gaseous species into a liquid system has been applied for numerous processes [1]. Some of the important applications include: (1) solubilizing gaseous species into a liquid system (i.e., carbonated drinks and fish pond aeration); (2) sweeping other soluble gases (i.e., O_2_ sweeping in microalgae cultivation); and (3) improving the flow of a highly viscous liquid (i.e., enhanced oil recovery). However, low gas–liquid mass transfer is the major bottleneck for this process. Imposing high aeration rates would normally overcome the mass-transfer limitation. However, a high aeration rate is strongly associated with the formation of large bubbles, leading to a low interfacial area due to the combination of high gas diffuser surface interaction and aeration velocity. Large bubbles and a high rate of bubble formation also cause the bubbles to travel rapidly, which limits the time for gas–liquid contact, hence lowering the overall mass transfer yield [2]. The slow contact time also limits its potential in niche applications such as microalgae cultivation and carbonated drink preparation. Therefore, controlling the diffuser material’s properties, specifically its wettability and pore size, is crucial to achieve a high mass transfer rate [3].

Membrane technology has matured over the years and has been applied commercially in industry. Membrane diffusers can be divided into rigid plate diffusers and perforated flexible rubber diffusers. The most commonly applied air diffuser in industry is the flexible rubber membrane due to its ability to self-clean during aeration, fewer clogging problems and uniform bubble formation [4,5]. In order to utilize the potential of a membrane air sparger, extensive research has been conducted on methods to control membrane pore properties and surface chemistry [6,7,8]. The potential of a membrane as an air diffuser has been explored in our previous work [9]. Microporous (MF) membranes as air diffusers can produce smaller bubbles (mean diameter = 3 mm) compared to the conventional bubble diffuser from a perforated tube (mean diameter of 12 mm). The decrease in the size of air bubbles in microalgae cultivation resulted in a higher biomass concentration (*Staurastrum* sp.) of 325 mg/L, 41% higher than that of the conventional devices over the same cultivation period. It was postulated that the smaller bubbles produced a higher liquid–gas contact area, and thus improved the CO_2_ (used as an inorganic carbon source) mass transfer to the microalgae culture. Therefore, the use of an MF membrane shows the potential to assist in microalgae cultivation by providing an improved CO_2_ mass transfer rate.

In theory, the formation of bubbles follows three steps: (1) As gas flows across the orifice, bubbles begin to grow due to the balance in the sum of resistance, consisting of hydrostatic pressure, surface tension and elastic pressure. (2) As the pressure difference inside and outside the bubble increases, the bubble surface grows larger. (3) As the bubble size keeps growing, the bubble neck starts to form and the bubble detaches itself form the orifice [10]. Among the factors affecting bubble formation, the hydrophobicity of the sparger’s surface plays a major role. Recently, an extensive study on the role of surface chemistry in bubble formation in air–water systems has been reported by Wesley et al. [3]. Using a coated sintered steel plate with a single 250 μm pore, two distinctive sizes of bubble formation were observed in hydrophobic and hydrophilic regions. The hydrophobic region (contact angle >90 degree) produced bubbles 5–6 mm in diameter, whereas the hydrophilic region (contact angle <90 degree) produced bubbles 2.5–3.0 mm in diameter. For the hydrophobic surface, the air tends to attach to the surface of the porous diffuser through hydrophobic–hydrophobic interactions, allowing the bubbles to grow bigger until buoyancy forces overcome the air-diffuser’s surface tension. In contrast, the interaction between the air and the hydrophilic surface of the pore is weaker, allowing smaller bubbles to have sufficient buoyancy to overcome the surface chemistry.

While bubble formation using a single pore diffuser has been well studied [11,12,13], there are a lack of studies on highly porous air spargers (i.e., porous membranes). Moreover, while membrane diffusers can produce smaller bubbles to improve the CO_2_ mass transfer area compared to conventional sparging or using a direct gas supply, the effect of the hydrophilic surface of the membrane needs to be addressed to further reduce the sizes of the bubbles formed. In this study, the effect of a PEBAX coating on the MF substrate, via vacuum filtration, on the bubble size is investigated at different PEBAX concentrations and air flow rates. To further study the effect of bubble size, hydrophilic and hydrophobic membranes were also chosen for *C. vulgaris* growth.

## 2. Materials and Methods

### 2.1. Materials

A polytetrafluoroethylene (PTFE, LSN#FBM142PTFE045H) membrane with a pore diameter of 0.45 μm was purchased from Merck (Darmstadt, Germany). It was used as a base hydrophobic membrane substrate. Polyether block amide (PEBAX, product code: A07381) was purchased from Arkema Inc. (Colombes, France) and was used as the hydrophilic polymer for the PTFE surface modification. This copolymer contains highly polar groups as a part of the polymer backbone. It is known to exhibit hydrophilic properties [14]. Ethanol (EtOH) with 95% purity was used as the solvent for PEBAX. All chemicals were used as received without prior purification. *Chlorella vulgaris* microalgae (*C. vulgaris*) and nutrients were obtained from the Centre of Biofuel and Biochemical Research (CBBR) of Universiti Teknologi PETRONAS (UTP, Perak, Malaysia).

### 2.2. PEBAX Coating

Firstly, a predetermined amount of PEBAX pellets were added to EtOH/H_2_O (mass ratio of 70:30) in a 100 mL Schott bottle to prepare PEBAX solutions of different concentrations (0.1 to 1.5 wt%). The mixture was then stirred using a magnetic stirrer at a temperature of 75 °C and at stirring speed of 200 rpm until a homogenous solution was obtained. The solution was then kept at ambient temperature prior to being used for coating.

For the PEBAX coating process, a PTFE membrane was first placed on a vacuum suction filter and 50 g of the prepared PEBAX solution was poured onto the membrane surface while the vacuum pump was running. Vacuum filtration was continued for 5 min to ensure that all the solution was filtered through the PTFE membrane. The coated membrane was then dried in an oven for 1 h at 70 °C. These steps were repeated for all the PEBAX solutions of different concentrations. The sample abbreviations in this report are based on the PEBAX concentrations. For example, PEBAX1.5 represents the PTFE membrane coated with PEBAX with 1.5 wt%.

### 2.3. Sample Characterization

ATR-IR analysis was carried out to verify the coating of the PEBAX layer and also to identify the presence of functional groups on the coated samples using a Frontier 01 Perkin Elmer spectrometer (Waltham, MA, USA). The surface morphologies of the prepared membranes were studied using a Zeiss EVO Scanning Electron Microscope (Oberkochen, Germany). The prepared samples were first coated with a thin film of gold by means of a sputter coater to avoid an overcharging effect. The SEM images were further subjected to ImageJ analysis to obtain the surface pore size and surface porosity, as well as pore-mouth geometry according to a method detailed elsewhere [15].

The membrane surface water contact angle (CA) was measured with a Contact Angle System OCA (NJ, USA). The membrane sample (1 cm × 1 cm) was adhered to a flat sample platform and a distilled water droplet (0.5 to 10 mm) was placed on it. An image of the droplet on the membrane was captured and the CA was determined by the SCA 20 software. Measurements were taken from five different positions to minimize experimental error and the average value was reported.

### 2.4. Bubble Size Determination

Bubbles were formed using a custom-built system (Figure 1). The MF membrane sample was cut to a diameter of 2.5 cm, and was placed in the system, sandwiched between two O-rings to avoid leakage. The cylinder above the membrane was filled with distilled water to a constant height of 25 cm. It was set low enough to minimize the impact of hydrostatic pressure against the flow direction of the air. The inlet air was injected at set flow rates of 0.1–50 mL/s by controlling a screw valve. At each flow rate, the formation of bubbles was recorded using 120 fps video, obtaining at least 10 different screenshots. The images were then analyzed using ImageJ^®^ to evaluate the cross-sectional area of the bubble. The equivalent diameter of the bubble was then calculated by assuming that the cross-sectional area of the bubble can be represented as a perfect circle of the equal cross-sectional area. The bubble formation frequency (number of bubbles produced per second, $\dot{n}$) and the effective total gas–liquid contact area per time (${\dot{A}}_{T}$) were then calculated using Equations (1) and (2), respectively.
(1)$$\dot{n}=\frac{\dot{Q}}{V_{bubble}}=\frac{6\dot{Q}}{\pi d_{B}{}^{3}}$$
(2)$${\dot{A}}_{T}=\dot{n}A_{B}=\frac{\dot{n}\pi d_{B}{}^{2}}{2}$$
where *d_B_* is the equivalent diameter of the bubble generated (cm), $\dot{Q}$ is the flow rate measured, and *A_B_* is the effective gas–liquid contact area of a single bubble (cm^2^) under the assumption that only top half of the bubble is effectively in contact with the liquid media.

### 2.5. Microalgae Cultivation

For *C. vulgaris* microalgae cultivation, a similar setup was used (Figure 1), using an algae cultivation medium instead of water, with PEBAX0 and PEBAX1.5 as membrane samples due to their apparent bubble properties. The algae cultivation conditions were adapted from the literature with minor modifications [16]. The membrane samples were cut to have diameters of 2.1 cm and sandwiched between the two O-rings with diameters of 5 cm. Each system was filled with 40 mL of microalgae cells with a predetermined amount of nutrients. The medium culture was supplied with CO_2_ by using an air pump supplied through the bottom of the system. The outlet flow rate was set at a constant 1.7 mL/s to control the bubble flow rate in the microalgae medium. The microalgae cultivation conditions were kept constant at a room temperature of 25 °C and a culture medium pH in the range of 3.0–3.5. The setup was illuminated with fluorescent light for 12 days. For analysis, a small amount of microalgae broth sample was taken each day of the 12-day cultivation period. The microalgae sample was then transferred into the cuvette for UV-VIS spectrophotometer (Shimadzu UV 1800, Kyoto, Japan) analysis. The absorbance of the microalgae sample was recorded and plotted on a graph.

## 3. Results and Discussion

### 3.1. Sample Characterization

The FTIR spectra for the samples of PEBAX0 to PEBAX0.5 are shown in Figure 2. The presence of dual C-F stretching at the wavelength of approximately 1100–1250 cm^−1^ originated from the PTFE polymer, the main building block of the membrane matrix. The absorption bands at the wavelengths of 1550 cm^−1^, 1650 cm^−1^, 2870–2940 cm^−1^ and 3300 cm^−1^ indicate the presence of H-N-C=O, C=O, C-H and N-H of the PEBAX, respectively. The intensity of the above three absorption bands increases along with the PEBAX concentration, which indicates the successful coating of the PEBAX layer at the different amounts.

The surface morphologies of samples PEBAX0, PEBAX0.5, PEBAX1.0, PEBAX1.25, and PEBAX1.5 are presented in Figure 3. The pore diameter of PEBAX0 is 0.46 μm, which is comparable to the specification provided by the manufacturer. The pore size of the membrane after coating was expected to be smaller as the coating solution would adhere to the PTFE pores and thus reduce their diameters. However, the PTFE pores were overlaid by the PEBAX layer, thus creating different pore characteristics while reducing the membrane porosity. Based on the smallest pore diameter, the pore size and porosity decrease as the PEBAX concentration increases due to the higher pore capillary pressure required for the PEBAX solution to enter into the pores. The pore sizes of PEBAX0.5, 1.0, 1.25 and 1.5% are 0.36 μm, 0.33 μm, 0.29 μm and 0.25 μm, respectively.

Figure 4 shows the CA values for the prepared membranes. The water CA for PEBAX0 is 118.45° ± 0.09°, which indicates a hydrophobic surface. Upon PEBAX coating, the membrane surface became hydrophilic even at a low PEBAX concentration; i.e., the CA of PEBAX0.1 is 47.59° ± 0.08°. This is due to the presence of functional polar groups (N-H and C=O (Figure 2)) that give the membrane surface a good affinity to water via hydrogen bonds. With an increasing PEBAX concentration (up to 1.5wt%), the CA values remain constant within the margin of error ±5°. It should be noted that there is no clear correlation between PEBAX concentration and CA.

### 3.2. Bubble Size Formation in the Membrane Bubble Diffuser

The study on bubble size formation was carried out for all prepared samples at different flow rates (Figure 5) and the results are summarized in Figure 6. At a low air flow rate, PEBAX0 produced a bubble with a diameter of 0.88 ± 0.05 cm (at an air flow rate of 0.25 mL/s). On the hydrophobic surface (θ > 90°), the air possesses a higher affinity to the membrane surface than water. This provides a strong air–membrane adhesive force that requires strong buoyancy forces to overcome. Hence, the bubble enlarges until the buoyancy force becomes strong enough for the bubble to escape from the surface, resulting in larger bubbles [3].

The presence of the PEBAX coating resulted in the reduction in the equivalent bubble diameter, even at the lowest concentration of 0.1 wt% PEBAX (Figure 4). In contrast with the hydrophobic surface, the hydrophilic surface caused low adhesive forces between the air and membrane and created bubbles with a significantly lower size. This behavior is illustrated in Figure 7.

It was also observed that increasing the PEBAX concentration produces smaller bubble size, where PEBAX1.5 produces bubbles with a diameter of 0.43 ± 0.01 cm (at an air flow rate of 0.59 mL/s). This trend agrees with the reduction in pore size and porosity as observed in Figure 3 and also agrees with the observations reported in the literature, where diffusers with smaller pores produce smaller bubbles [12,17,18].

Figure 6 shows that the equivalent bubble diameter increases with an increasing air flow rate. This finding can be attributed to bubble elongation which increases the time period for the bubble to detach from the surface. This is also known as the bubble departing period, *τ*, which promotes the formation of larger bubbles [19]. Similar phenomena occur regardless of the level of the surface hydrophilicity. However, with a more hydrophilic surface, *τ* is shortened due to its weaker adhesion to the surface compared to a more hydrophobic surface at the same air flow rate. A higher air flow rate would also promote bubble coalescence to form a larger bubble size. These phenomena are discussed in detail later in the paper.

It should be noted that the bubble diameters produced in this work are larger than the values reported in the literature, where hydrophilic and hydrophobic surfaces are also studied and compared [3,18]. The work by Wesley et al. [3] focuses on very low flow rates (0.04–0.92 mL/s), producing bubbles 2.5–3.0 mm in diameter, whereas the work by Kukizaki et al. [18] involves the sweeping of liquid to detach bubbles from the surface, resulting in an average bubble size of 8 μm. This work focuses on a relatively higher flow rate (0.07–50 mL/s) without the sweeping liquid. The findings in this work therefore fill the knowledge gap currently lacking in the literature.

In general, the bubble formation frequency of the hydrophilic coated membranes is significantly higher than the hydrophobic uncoated surface (Figure 8); i.e., the rate of bubble formation by PEBAX1.5 is 5.5 times higher than PEBAX0 at the same air flow rate of 43 mL/s. These results are expected, since under the same flow rate, more bubbles are formed to compensate for the reduction in bubble diameter.

It was also observed that the rate of bubble formation peaks for a given flow rate, followed by relatively constant values at high flow rates. This finding is in accordance with the observations made by Thorns et al. [20]. Their results are more clear for hydrophilic surfaces, particularly due to the smaller bubble size produced (Figure 7). To further elaborate on this behavior, the equivalent bubble diameter and bubble formation frequency at different air flow rates for PEBAX1.5 were replotted (Figure 9). According to Zhang and Shoji [19], bubble formation at different air flow rates can be classified into four regimes: single bubble, pairing, double coalescence, and triple coalescence. A similar behavior was also observed in this study. At a low air flow rate (Range I), a single bubble is formed without being influenced by a previously formed bubble [21]. At a medium-low flow rate (Range II), the vertical elongation of the previously formed bubble promotes the following bubble’s formation without coalescence due to a sudden pressure drop as the bubble is formed; hence, bubble pairing is observed. A further increase in the flow rate (Range III) results in two successive bubbles coalescing, causing a sudden increase in the equivalent bubble diameter. The coalescence reduces the number of bubbles formed and a sudden drop in bubble formation frequency is observed (Figure 9). At a high air flow rate (Range IV), multiple coalescences between two or more bubbles occur, forming substantially larger bubbles. Consequently, bubble diameter increases linearly with increasing air flow rates.

The total contact area per unit of time, ${\dot{A}}_{T}$, can be estimated and the results are summarized in Figure 10. Overall, the bubble contact area increases as bubble size decreases (higher PEBAX concentration) as smaller bubbles produce a higher effective interfacial area per unit volume. PEBAX1.5 shows the most prominent result compared to other membranes. It increases the overall effective interfacial area and proves that a hydrophilic surface produces smaller and more bubbles which then increases the overall effective interfacial area (Figure 6 and Figure 7). Regarding the effect of air flow rate, the total bubble contact area also increases with the air flow rate. With a medium air flow rate, e.g., at 10 mL/s, PEBAX1.5 has a total bubble contact area of 37.35 cm^2^/s, which is almost double that of PEBAX0 (18.76 cm^2^/s). At higher flow rates, e.g., at 46.67 mL/s, PEBAX1.5 has a total bubble contact area of 91.33 cm^2^/s, which is only 77% higher than PEBAX0 (51.34 cm^2^/s), due to a higher rate of bubble coalescence at a higher flow rate. Therefore, it can be concluded that a hydrophilic surface yields a higher effective interfacial area and the effect becomes prominent with an increasing air flow rate. It should also be noted that the effective interfacial area is also affected by the increase in the equivalent bubble diameter and increased bubble formation frequency as the air flow rate increases (Figure 9).

### 3.3. Microalgae Cultivation

To further study the effect of bubble size, PEBAX0 and PEBAX1.5 were used for the *C. vulgaris* cultivation. Figure 11 shows the biomass concentration throughout the 12 days of cultivation. Overall, it can be observed that biomass concentration for PEBAX1.5 (with average bubble size of 0.43 ± 0.07 mm) shows an increment as much as 112% higher than that of PEBAX0 (with average bubble size of 0.40 ± 0.05 mm) at day 12. This result is expected due to the smaller bubble size produced by PEBAX1.5 (Figure 6), which, compared to PEBAX0, increases the overall effective air–liquid interfacial area (Figure 10), as well as the air supply to the algae medium.

It was also noted that PEBAX1.5 could yield as much biomass concentration (a maximum of 1.3 g/L at D-12, supplied by air) as when 2% or 5% CO_2_ is directly supplied to the microalgae cell as reported elsewhere [22,23]. Previously, Chiu et al. reported that 2% CO_2_ in an aerated culture increased the biomass concentration to 1.2 g/L from 0.5 g/L when air was supplied to the culture [22]. Meanwhile, Lam and Lee found that 5% CO_2_ increased the biomass concentration up to 0.75 g/L, compared to only 0.5 g/L when only air is supplied [23]. Therefore, the use of a hydrophilic membrane as an air diffuser (especially PEBAX1.5) is an effective alternative to be applied for microalgae cultivation, as it improves the gas transfer rate in a gas–liquid medium.

## 4. Conclusions

The surface of a porous PTFE membrane was successfully coated via the vacuum filtration method, as confirmed by the FTIR, the SEM images and the CA values. The coating with PEBAX solutions (via vacuum filtration) of different concentrations resulted in a significant decrease in the contact angle, e.g., from 118° for PEBAX0 to 43.66° for PEBAX1.5. This improved hydrophilicity leads to the formation of smaller and more bubbles, and consequently to a larger gas/liquid interfacial area. Furthermore, PEBAX1.5 improves the CO_2_ mass transfer in a microalgae cultivation culture, because the biomass concentration increased by 112%, from 0.6 g/L to 1.30 g/L, at the end of the 12 days of cultivation. Overall, facile modifications via vacuum filtration have proven to be a promising technique to improve the gas–liquid mass transfer that could facilitate microalgae growth and can be applied for other applications. It should also be noted that the *C. vulgaris* cultivation was conducted to study the effect of bubble formation on the biomass concentration only to support our hypothesis earlier stated in Section 3.3, where hydrophilic membrane produces smaller bubbles with high effective interfacial area. Therefore, further study on membrane fouling during algae cultivation should be considered to improve the performance of PEBAX1.5 as a superior gas sparger.

## Figures and Tables

**Figure 1 membranes-12-00414-f001:**
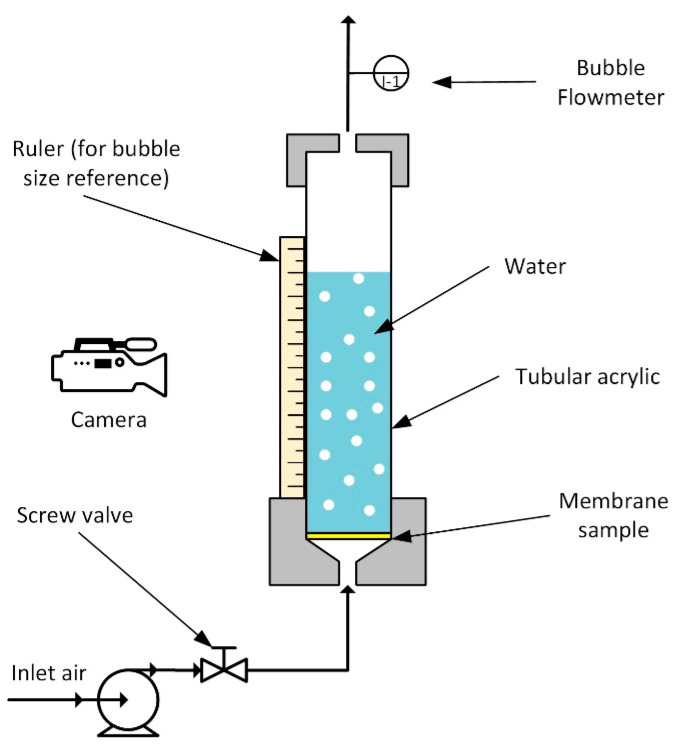
Custom experiment set-up.

**Figure 2 membranes-12-00414-f002:**
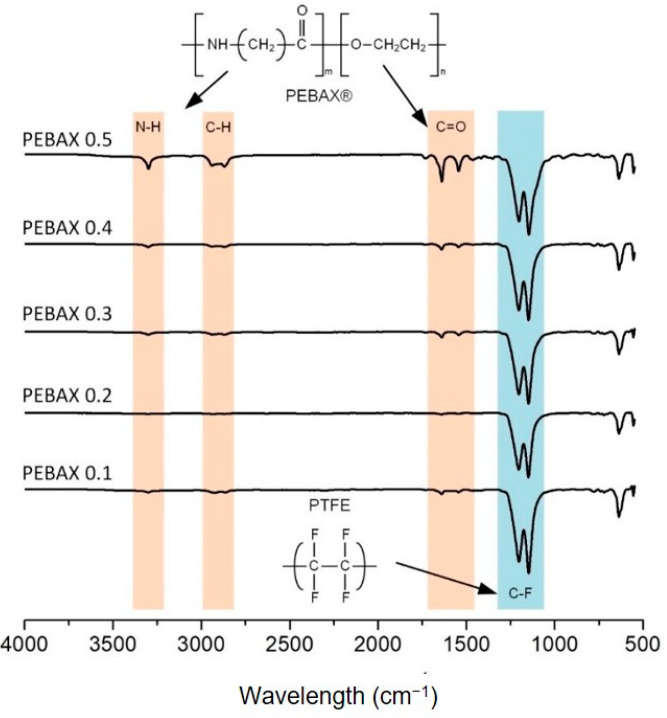
FTIR analysis of coated PTFE with PEBAX concentrations of 0.1–0.5 wt%.

**Figure 3 membranes-12-00414-f003:**
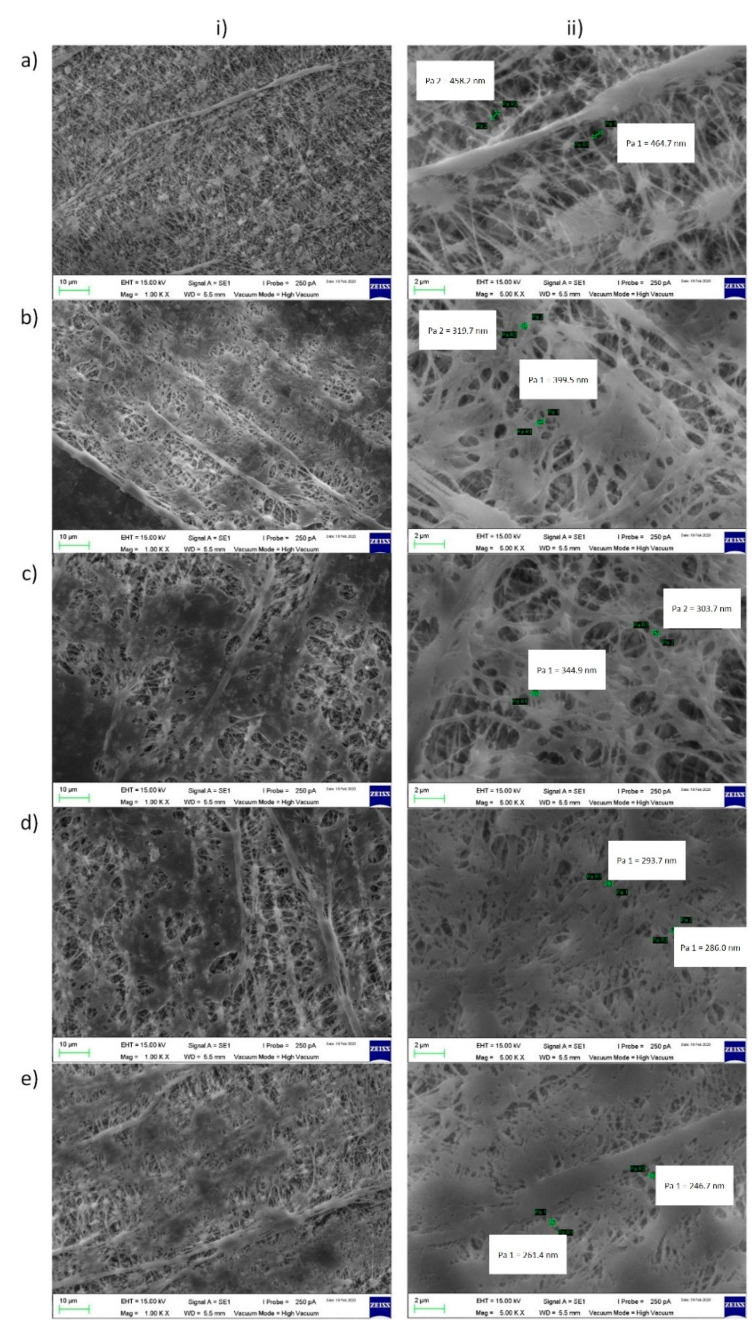
Surface morphologies of (**a**) PEBAX 0, (**b**) PEBAX 0.5, (**c**) PEBAX 1.0, (**d**) PEBAX 1.25, and (**e**) PEBAX 1.5, at (**i**) 1.0 k magnification and (**ii**) 5.0 k magnification.

**Figure 4 membranes-12-00414-f004:**
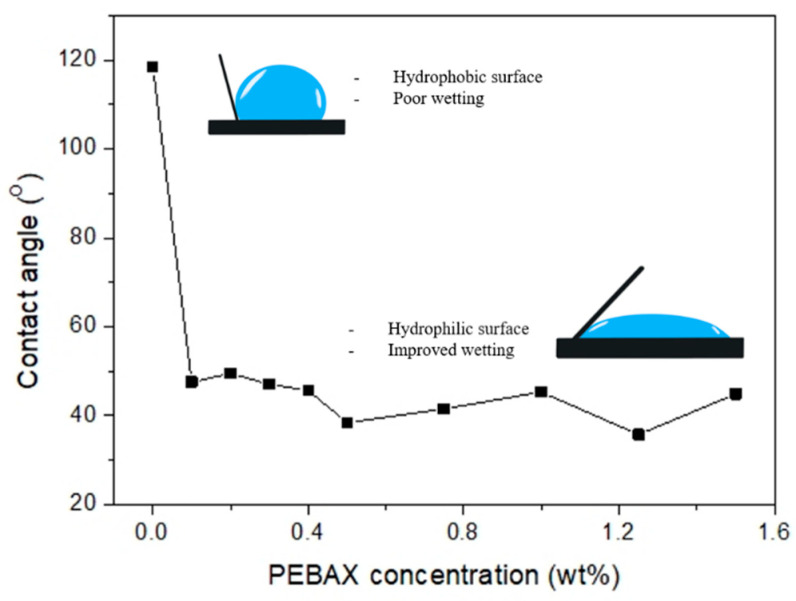
Effect of PEBAX concentration on contact angle.

**Figure 5 membranes-12-00414-f005:**
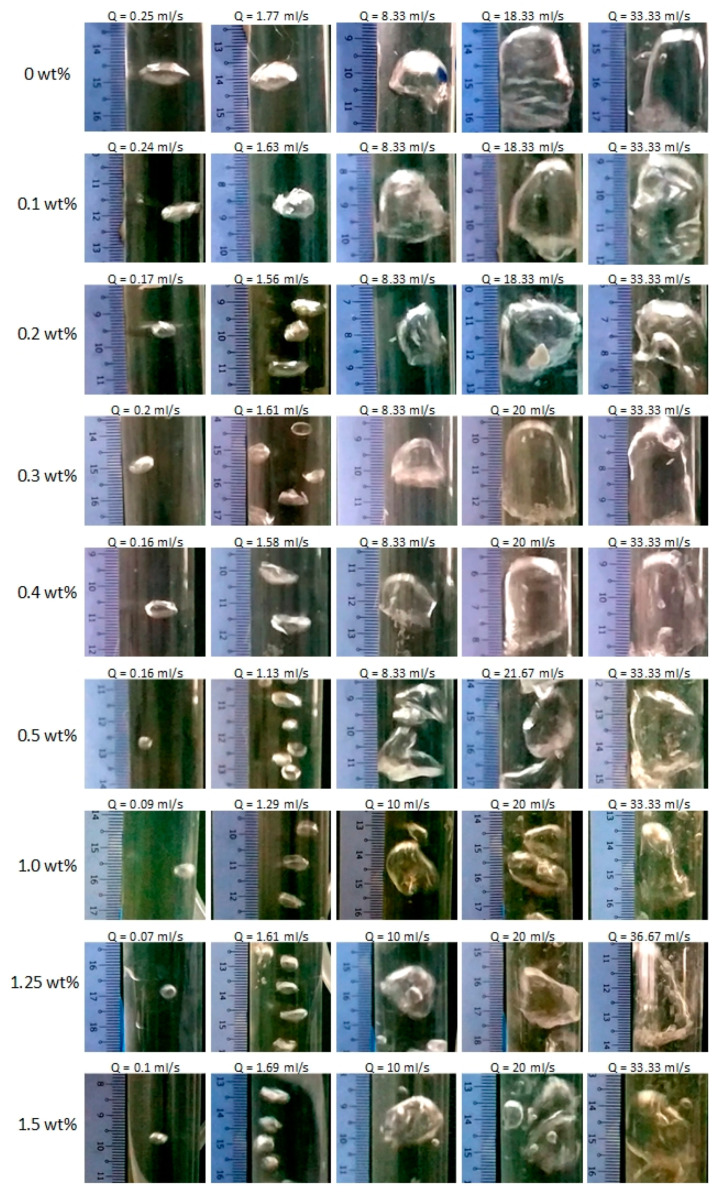
Bubble formations at different coating concentrations under variable air flow rates.

**Figure 6 membranes-12-00414-f006:**
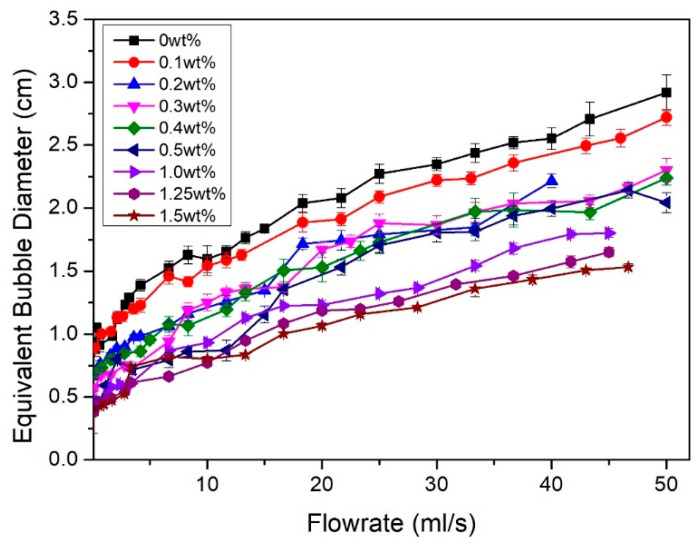
Equivalent bubble diameter of bubbles produced at different PEBAX concentration under different air flow rates.

**Figure 7 membranes-12-00414-f007:**
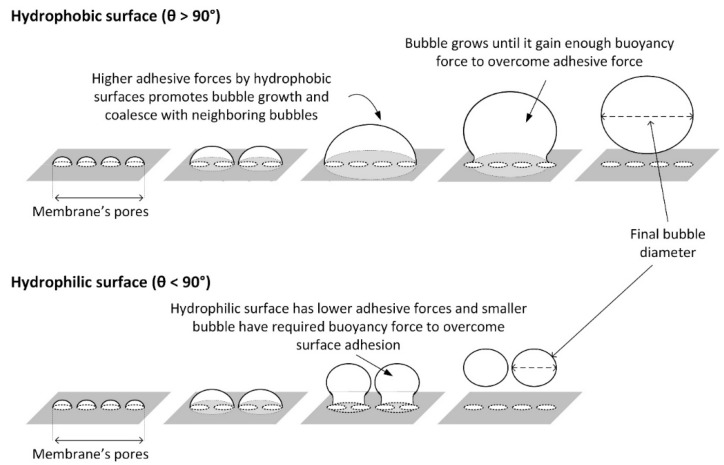
Bubble formation using hydrophobic (θ > 90°) and hydrophilic (θ < 90°) porous membrane surfaces.

**Figure 8 membranes-12-00414-f008:**
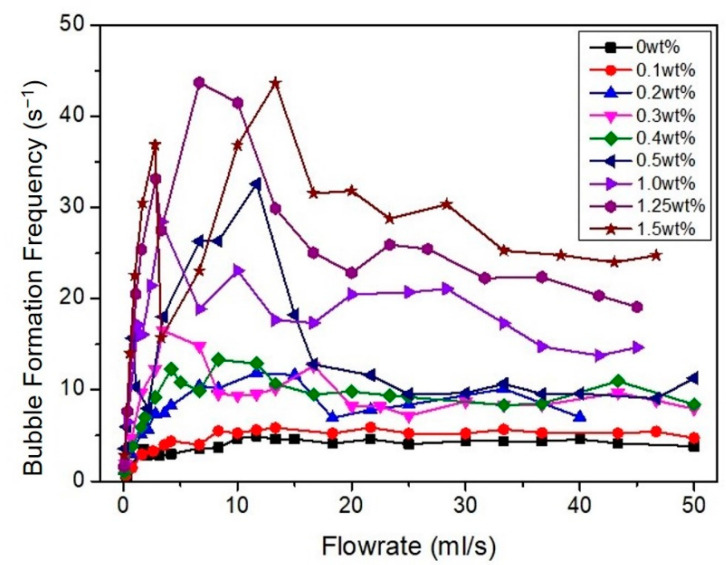
Bubble formation frequency (s^−1^) at different PEBAX concentrations under different air flow rates.

**Figure 9 membranes-12-00414-f009:**
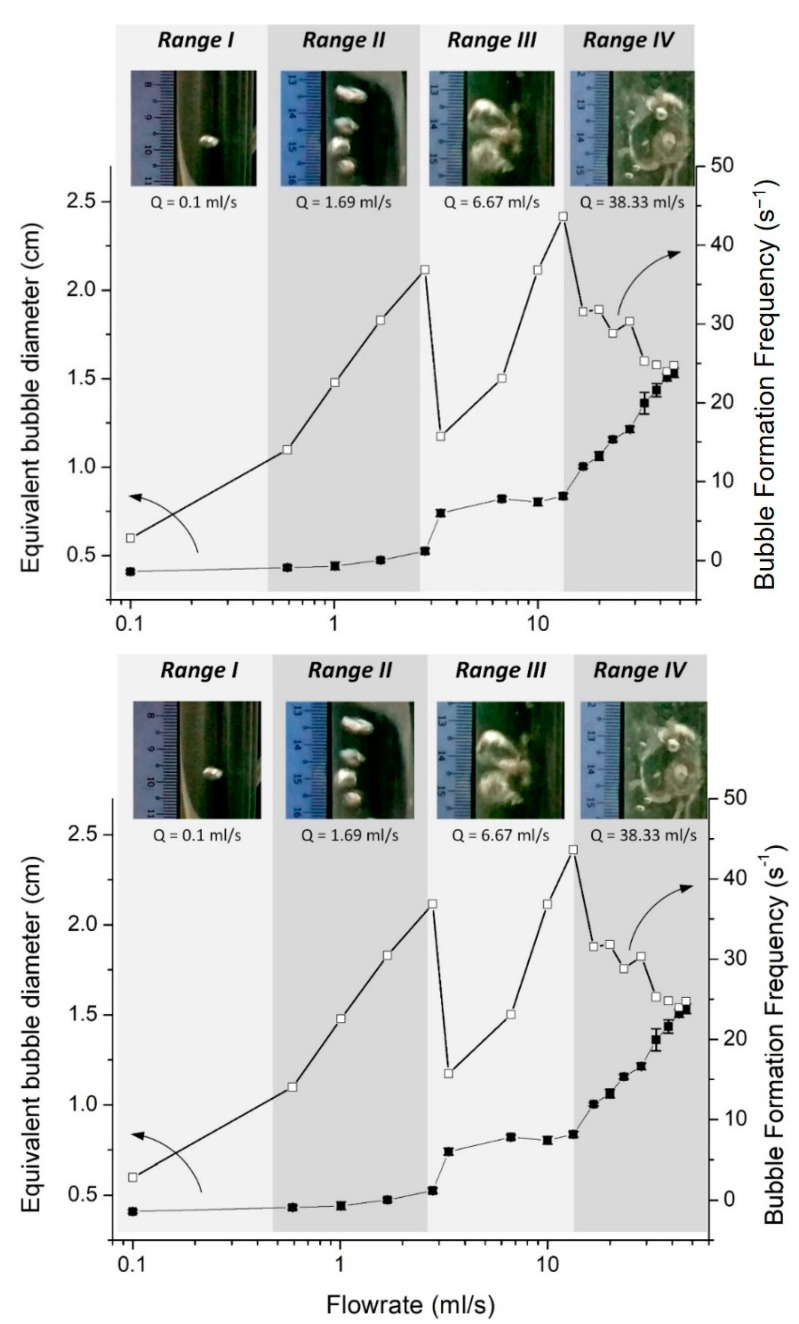
Schematic diagram of bubble formation at different air flow rates.

**Figure 10 membranes-12-00414-f010:**
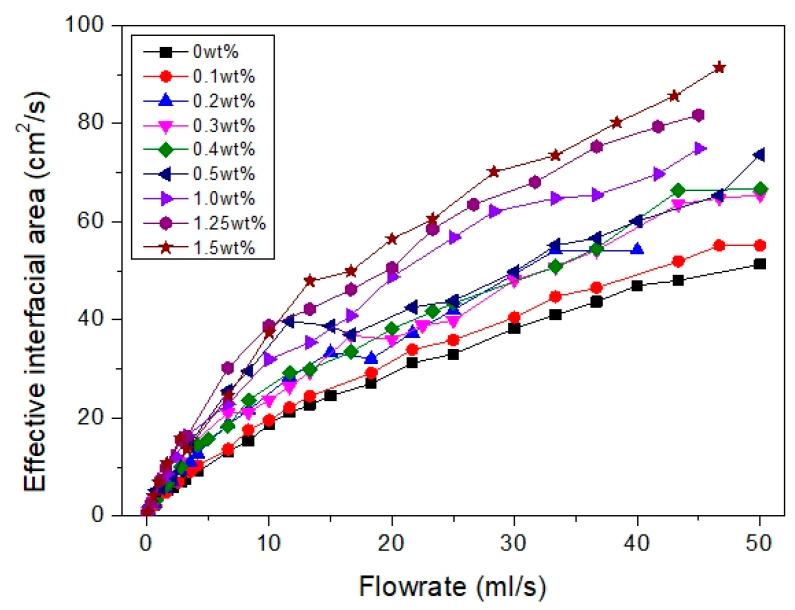
Effective air–liquid interfacial area at different PEBAX concentration under different air flow rates.

**Figure 11 membranes-12-00414-f011:**
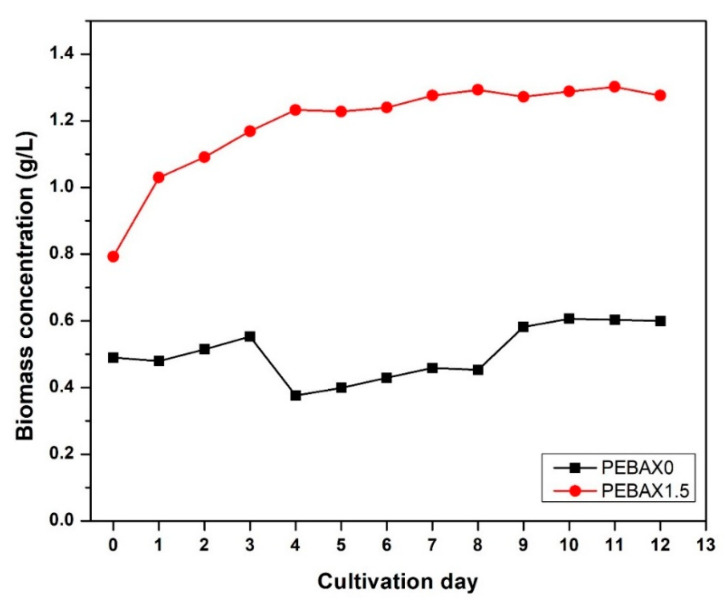
Biomass concentration (g/L) of microalgae for PEBAX0 and PEBAX1.5 throughout 12 days of cultivation.

## Data Availability

Not applicable.

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
