# Peer review of "Controlling Air Bubble Formation Using Hydrophilic Microfiltration Diffuser for C. vulgaris Cultivation"

_membranes, 2022, doi:10.3390/membranes12040414_

Round 1
Reviewer 1 Report
The reviewed manuscript, number membranes-1656103, titled: ‘Controlling Air Bubble Formation Using Hydrophilic Microfiltration Diffuser For C. Vulgaris Cultivation’ describes the preparation and characterization of PTFE membrane coated with hydrophilic layer of PEBAX and its potential application in the microalgae cultivation. The study seems to be very interesting. The manuscript is well written, the characterization methods were appropriate chosen, the results are clearly described and the conclusions are confirmed by the conducted research. In my opinion, the reviewed manuscript is worth to be published in Membranes after a minor revision. My comments are as follows:
- I couldn’t agree, that the absorption band at about 1550 cm-1 is attributed to C=O vibration. It should be corrected.
- In paragraph 3.1: it is better to use ‘absorption band’ than ‘peak’ regarding the signals present in the FTIR spectrum.
- Correct the spelling of ‘flow rate’, it is written ‘flowrate’ in many places in the manuscript.
- Paragraph 2.4 and Figure 11 are missing; please correct the numbering.
Reviewer 2 Report
The work is well planned and has interest in the field. However, I believe that the authors should take into account the following considerations.
Lines 95-96: They must give the commercial reference of the membrane they have used, not just the name of the manufacturer.
Line 129: Is this value of 100mm correct?
Lines 187-188: I believe that the authors should also report the variation in porosity. The contact angle is related to the porosity. In addition, porosity is likely to determine the size of the bubbles. According to the mechanism proposed by the authors in figure 7, I think that it is also important to know the mean distance between pores.
Lines 196-197: Figure 4 should be mentioned here. It can be seen that, within the margin of error, the contact angle remains constant for all the PEBAX-coated membranes. I think that they should mention this, and they should take it into account in their further analysis.
Lines 215-217: The authors say “This trend associates well with the reduction in pore size and porosity as observed in Figure 3 and also corroborates with the observation reported in the literature, in which diffusers with smaller pore produce smaller bubbles [12, 17, 18].” The authors have data on both mean pore size and porosity, obtained from SEM images. I think it is very interesting to analyze the behavior of the bubble size as a function of these data (porosity and pore size). As the authors show, the lowest concentration of PEBAX (0.1%wt) gives approximately the same hydrophilic character as 1.5%wt PEBAX. I think (and they also suggest) that the variation of bubble size with PEBAX concentration should be attributed primarily to pore size (and/or porosity). I believe that the treatment with PEBAX (0.1%) already marks the reduction in bubble size due to the increase in hydrophilicity.
Lines 287-289: The authors say “Therefore, it can be concluded that hydrophilic surface yields higher effective interfacial area where the effect becomes prominent with increasing air flow rate.” I still think that this effect is not due to hydrophilicity alone.
Line 297: Figure 11 does not exist.
Lines 313-315: Idem to what was said for lines 287-289.
Conclusions: I think they should look at the effects of pore size and porosity, and include these results in their conclusions. Reading the manuscript clearly indicates this dependency. The authors should study this effect, and include the results of this study in the conclusions.
Round 2
Reviewer 2 Report
I believe that the manuscript can be published in its current state. However, I think that some improvements (in addition to those already made) could have been made by the authors.